# Characterization of Zn-Mg-Sr Type Soldering Alloy and Study of Ultrasonic Soldering of SiC Ceramics and Cu-SiC Composite

**DOI:** 10.3390/ma16103795

**Published:** 2023-05-17

**Authors:** Roman Kolenak, Alexej Pluhar, Jaromir Drapala, Paulina Babincova, Matej Pasak

**Affiliations:** 1Faculty of Materials Science and Technology in Trnava, Slovak University of Technology, Jana Bottu No. 2781/25, 917 24 Trnava, Slovakia; roman.kolenak@stuba.sk (R.K.); paulina.babincova@stuba.sk (P.B.); matej.pasak@stuba.sk (M.P.); 2Faculty of Materials Science and Technology, Technical University of Ostrava, 17 Listopadu 2172/15, 708 00 Ostrava, Czech Republic; jaromir.drapala@vsb.cz

**Keywords:** soldering, SiC ceramics, Cu-SiC composite, Zn-Mg-Sr solder, ultrasonic soldering

## Abstract

The aim of the research was to characterize the soldering alloy type Zn-Mg-Sr and direct the soldering of SiC ceramics with Cu-SiC-based composite. It was investigated whether the proposed composition of the soldering alloy was appropriate for soldering those materials at the defined conditions. For the determination of the solder melting point, TG/DTA analysis was applied. The Zn-Mg system is of the eutectic type with a reaction temperature of 364 °C. The effect of strontium on the phase transformation was minimal, owing to its lower content. The microstructure of the soldering alloy type Zn3Mg1.5Sr is formed of a very fine eutectic matrix containing segregated phases of strontium—SrZn_13_ and magnesium—MgZn_2_ and Mg_2_Zn_11_. The average tensile strength of the solder is 98.6 MPa. The tensile strength was partially increased by solder alloying with magnesium and strontium. The SiC/solder joint was formed due to the distribution of magnesium from the solder to the boundary with the ceramics at the formation of a phase. Owing to soldering in air, oxidation of the magnesium took place and the oxides formed were combined with the silicon oxides that remained on the surface of the ceramic material—SiC. Thus, a strong bond based on oxygen was obtained. An interaction between the liquid zinc solder and the copper matrix of the composite substrate took place at the formation of a new phase—γCu (Cu_5_Zn_8_). The shear strength was measured on several ceramic materials. The average shear strength of the combined SiC/Cu-SiC joint fabricated with Zn3Mg1.5Sr solder was 62 MPa. When soldering similar ceramic materials mutually, a shear strength of as much as around 100 MPa was observed.

## 1. Introduction

Pb-Sn-based solders with a high proportion of lead (80–95 wt. %) were for some time the solders most used in the service conditions of semi-conductor power modules. These solders demonstrated excellent properties, such as a suitable melting point, relatively low price, good wettability on metal coatings, and acceptable mechanical properties [1,2]. However, on 1 July 2006, the EU guideline WEEE (Waste from Electrical and Electronic Equipment) introduced a rule in order to restrict the volume of waste from electrical and electronic equipment [3,4]. Since lead is included in this as well, there was an inevitable need to switch to lead-free solders, which has raised considerable issues in technical practice. Extensive research is in progress at present, aiming to find replacements for high-lead solders for higher application temperatures that have comparable properties in terms of their quality and reliability. Several potential alloy systems exist that approach these requirements. These are the solders based on gold, silver, or bismuth, but their high price considerably limits their mass use [5,6,7,8,9,10]. Therefore, the most suitable candidate seems to be zinc-based solders, which, apart from a favorable price, also offer high strength, good thermal and electrical conductivity, and high reliability. Currently, available literary sources mainly mention the Zn-Al-based solders, which have a suitable melting range, high corrosion resistance, and excellent mechanical properties, making them suitable candidates for use in the production of power semiconductors, optical devices, and other electronic applications [11,12,13].

The authors of the research [14,15] focused on the corrosion and mechanical properties of biodegradable Zn3Mg0.7Y and Mg-Ca-Zn alloys. The authors discovered that the poor mechanical properties of Zn could be significantly improved by the addition of Mg and Y, which was confirmed in the above studies.

Meanwhile, the new soldering alloys are increasingly being designed with their versatility in mind, meaning their ability to join direct combinations of materials, such as metal/ceramics or metal/composites. For such purposes, these so-called active solders are being designed nowadays, alloyed with an active element and able (under certain conditions) to create a bond also with ceramic materials, semiconductors, or glass materials. The effects of the active elements in the solders destined for soldering diverse materials are well documented in our previous works [16,17,18]. In those works, we focused on the joining of diverse ceramic materials with metals by applying a progressive hybrid soldering technology, namely hot plate/ultrasound.

Research on soldering ceramic or metallic material with Zn-based solders is continuing. Xiao et al. [19,20] soldered Cu with Cu and Cu with Al using Zn–3Al solder with ultrasound assistance. The authors declared that the Zn–5Al solder is a better option for soldering SiC ceramics since it has a lower melting point (381 °C). However, the massive amount of fragile Zn-Al eutectics necessarily degrades the mechanical properties of the joints. Therefore, the microstructure needs to be optimized via the design of the alloy composition or by optimizing the soldering process. Optimization of the microstructure within this context was realized in the work of W.B. Guo et al. [21], who found that the addition of Cu to Zn-Al solder can lead to structural modification by eliminating the fragile eutectic Zn-Al phase, and to the enhancement of the mechanical properties of the joints. Zinc solders containing magnesium are another promising candidate. Research on the direct soldering of SiC ceramics using Zn-Al-Mg solder was published in the work of Chen et al. [22], who revealed the presence of a new shapeless layer of SiO_2_, 2 to 6 μm wide, on the SiC/solder boundary. The atoms of the eroded SiO_2_ layer diffuse fast to the SiC substrate due to the jet effect produced by ultrasound. The strong bond between the SiC substrate and the Zn-Al-Mg solder is attributed to the transfer of SiO_2_ mass to the solder induced by cavitation erosion. One of the advantages of using zinc solders alloyed with aluminum and manganese lies in zinc’s ability to form a eutectic structure with magnesium and thus prevent the formation of brittle intermetallic Al-Mg phases [23]. In order to avoid the residual potential of the formation of these brittle phases, solders without aluminum, based on Zn-Mg, are being designed. This base seems to be another suitable candidate for applications serving at elevated temperatures. In our previous work [24], we have dealt with the research on soldering Al-Al_2_O_3_ composite with copper by applying a Zn-Mg-In solder type. Soldering was realized in the air without flux by applying a power ultrasound. Magnesium and indium in the solder were used as the active elements supporting the reaction with the ceramic component of the composite. Joints free from defects or any inhomogeneities were achieved. However, the low strength of the joints was caused by the indium, which had distributed to the grain boundaries during cooling down.

Thus, in order to attain higher strength, indium must be eliminated from the soldering alloy composition. It may be replaced by an active element with a higher affinity to the surface of the ceramic materials. Therefore, this study is devoted to the characterization of a Zn-Mg-Sr-based solder and to its feasibility for the soldering of SiC ceramics and the progressive metal-ceramic material of Cu-SiC. It was investigated whether the proposed composition of the soldering alloy was appropriate for soldering those materials at the defined conditions. The investigation thus consisted of the study of both the solder itself and the interactions on the solder/substrate boundary.

## 2. Materials and Methods

After the determination of the mass proportions of the prepared alloy, the individual components were weighed. As the input components for solder manufacture, materials with high purity of 4N were used. Manufacturing of the solder in the as-cast condition was realized in an induction oven. The procedure was as follows: the charge was placed in a cold crucible; the zinc was heated at 700 °C and subsequently melted down until full fusion; magnesium and strontium were wrapped in a Zn foil and immersed in the Zn melt, then stirred with a ceramic stick until full fusion; the melt was then cast in a graphite crucible; and casting was performed without a shielding atmosphere.

The chemical composition given in Table 1 was prepared experimentally. Chemical analysis of the alloys was performed via atomic emission spectrometry with induction coupled plasma (ICP-AES) on the instrument type SPECTRO VISION EOP. The specimens of alloys for ICP-AES analysis were dissolved in suitable chemical solutions of acids and bases. The analysis proper was performed on an emission atomic spectrometer with a pneumatic atomizer and a Scott sputtering chamber.

Next, a test sample for tensile testing was prepared from the soldering alloy. A test piece of solder used for the static tensile test is mentioned in a research article from Kolenak et al. [25].

Substrates of the following materials were used in the experiments:ceramic SiC substrates in the form of discs Ø 15 × 3 mm,metal-ceramic Cu-SiC substrate with 4N purity and dimensions of Ø 15 × 3 mm and 10 × 10 × 3 mm.

The scheme of the soldered joint prepared for the chemical analysis of the solder/substrate boundaries is shown in Figure 1a.

The joints were made by using a hot plate with thermostatic control. The SiC substrate was laid on the hot plate and the solder was deposited on it. This assembly was then heated at the soldering temperature. The soldering process was carried out by using the ultrasonic equipment type Hanuz UT2 with the parameters given in Table 2. Activation of the solder was realized through an encapsulated ultrasonic transducer composed of a piezo-electric oscillating system and a Ti sonotrode with the end tip dimension Ø 3 mm. The temperature of the soldering process was 150 °C and was checked by a continuous temperature measurement on the hot plate using a NiCr/NiSi thermocouple. The ultrasound power was activated for 5 s. Soldering was performed without flux. After ultrasound activation, the redundant layer of oxides on the surface of the molten solder was removed. The same process was also performed with the second substrate. Then, both substrates with molten solder were laid on each other and the joint was thus achieved. The schematic representation of this procedure is shown in Figure 2.

The metallographic preparation of specimens from the soldered joints was realized by standard metallographic procedures. Grinding was performed using SiC emery papers with 240, 320, and 1200 grains/cm^2^ granularity. Polishing was performed with diamond suspensions with grain sizes 9 μm, 6 μm, and 3 μm. Final polishing was performed using the polishing emulsion type OP-S (Struers) with 0.2 μm granularity.

The microstructure evaluation was performed by JEOL JSM 7600F scanning electron microscopy (SEM, Jeol Ltd., Tokyo, Japan) with a Schottky field emission electron source operating at 20 kV and 90 µA. The samples were placed at a working distance of 15 mm and documented using a backscattered electron detector. The chemical element analysis was performed via the Oxford Instruments X-Max silicon drift detector and the energy dispersive X-ray spectrometer (EDS, Oxford Instruments plc, Abingdon, UK).

The XRD analysis was carried out on tensile test specimens on a PANalytical Empyrean X-ray diffractometer (XRD) (Malvern Panalytical Ltd., Malvern, UK). The measurements were performed via Bragg–Brentano geometry. Theta-2Theta angle range between 10° and 140° 2Theta was chosen. The XRD source with a Co anode was set to 40 kV and 40 mA. The incident beam was modified by a 0.04 rad soller slit, 1/4° divergence slit, and 1/2° anti-scatter slit. The diffracted beam path was equipped with a 1/2° anti-scatter slit, 0.04 rad soller slit, Ni beta filter, and PIXcel3D position sensitive detector operated in 1D scanning mode. The phase quality was analyzed using the PANalytical Xpert High Score program (HighScore Plus version 3.0.5 developed by PANalytical BV in United Kingdom) with the ICSD FIZ Karlsruhe database.

The TG/DTA analysis of the Zn3Mg1.5Sr solder was performed on the equipment type DTA SETARAM Setsys 18TM. The measuring system was provided with a cylindrical oven with a graphite heating body and the corresponding control thermocouple, measuring bar, and cooling medium. The crucibles (Al_2_O_3_) in which the measuring and reference specimens were inserted were freely laid on the measuring bar and brought into contact with two thermocouples, which served for measuring the difference in temperature between the studied and reference specimens. Prior to each analysis, the inner space of the oven was flushed with argon (6N purity) for around 15 min, then vacuumed again and filled with argon. During the analysis proper, a stable dynamic atmosphere was maintained in the oven space (with the Ar flow rate of 2 L/h). The analyzed temperatures, when the specimen was heated at a rate of 4 °C/min from room temperature up to the temperature of full fusion of the specimen, were recorded and evaluated on a PC using the SetSoft program. From the evaluated DTA curves, the temperatures of the phase transformations were read in the liquidus-solidus range as well as the phase transformations in the solid state. Additionally, the enthalpies of the phase transformations in the zones of prominent peaks were determined.

For the determination of the mechanical properties of the soldered joints, a shear test was performed. The schematic representation of the specimen is shown in Figure 2b. The shear strength was determined on a versatile tearing equipment type LabTest 5.250SP1-VM. A soldering jig with a defined form of the test sample was applied in order to change the direction of the axial tensile force acting on the test specimen (Figure 3). This shearing jig ensures a uniform loading of the specimen by shear in the plane of the solder and substrate boundary.

## 3. Results

### 3.1. TG/DTA Analysis

The TG/DTA analysis was carried out two times in each case at the heating and cooling rate of 5 °C/min (Figure 4 and Figure 5). The impressive transformation temperatures in the soldering alloy type Zn3Mg1.5Sr obtained by the TG/DTA analysis are given in Table 3.

A significant peak can be seen in the TG/DTA graph, which accords with the eutectic reaction. The Zn-Mg system is of the eutectic type (Figure 6) with the eutectic reaction at a temperature of 364 °C. The eutectic point lies on the zinc side (92.2 at.% Zn). The composition of the Zn3Mg1.5Sr alloy is slightly over-eutectic, which is why it was not easy to determine precisely the liquidus temperature during the heating phase in the DTA analysis. The onset of crystallization is observable just on the cooling curves and is higher than the temperature of the eutectic reaction (364 °C). The effect of strontium on the phase transformation was minimal owing to its lower content.

### 3.2. Microstructure of Zn3Mg1.5Sr Solder

The microstructure of the Zn3Mg1.5Sr solder (Figure 7) is formed of a very fine eutectic matrix where the phases of strontium (light-grey constituents) and the magnesium phase (grey constituents) are segregated.

For the detection of the chemical structure of the individual components of the soldering alloy, EDX analysis was carried out. The measurement spectrums are given in Figure 8, marked by numerals ranging from one to ten. The results from this measurement are shown in Table 4.

The measurement points of Spectrums 1 and 2 are formed of very fine eutectics, composed of a solid solution (Zn) and an intermetallic compound—Mg_2_Zn_11_.

At the measurement points of Spectrums 3 and 4, large light-grey constituents of irregular shape occur, which are formed of the SrZn_13_ intermetallic phase (Figure 9).

The measurement points of Spectrums 5 and 6 show the rounded constituents containing pure zinc.

The measurement points of Spectrums 7 and 8 represent the dark-grey constituents, unambiguously corresponding to the intermetallic phase of MgZn_2_.

At Spectrum points 9 and 10, the Mg_2_Zn_11_ phase occurs and it is located between the MgZn_2_ phase and pure Zn.

The diffraction XRD analysis of the Zn3Mg1.5Sr solder proved, similarly to the EDX analysis, the existence of a solid solution of zinc (Zn); intermetallic phases of magnesium, namely MgZn_2_ and Mg_2_Zn_11_; and the intermetallic phase of strontium—SrZn_13_. The report from the diffraction analysis is shown in Figure 10.

The planar distribution of the magnesium and strontium phases in the eutectic zinc matrix is documented in Figure 11. The strontium phases are just faintly indicated in Figure 11d since they contain a great amount of zinc, up to 90 wt. %.

### 3.3. Tensile Strength Test of Soldering Alloy

The tensile strength tests were intended to investigate the tensile strength of the Zn3Mg1.5Sr alloy. The parameters of the test specimens were computed. Three specimens were used for the measurement tests of the tensile strength of the soldering alloy. The loading rate of specimens was 1 mm·min^−^^1^. The average tensile strength was 98.6 MPa.

### 3.4. Microstructure of SiC/Zn3Mg1.5Sr/Cu-SiC Joint

The soldered joint of SiC/Zn3Mg1.5Sr/Cu-SiC was prepared at a temperature of 380 °C. Due to the ultrasound activation, a suitable joint was obtained in the soldering process, containing no cracks or inhomogeneities. The microstructure of the soldered joint is shown in Figure 12.

From Figure 12, it is evident that no significant strontium or magnesium phases crop up in the solder matrix. The distribution of Mg, Zn, and Sr elements after soldering is documented on the map in Figure 13. The strontium in Figure 13d shows an approximately similar distribution after soldering in the solder matrix as well.

### 3.5. Analysis of Transition Zone of SiC/Zn3Mg1.5Sr Joint

At the SiC/solder transition zone, a corrugated interface with depressions was formed by the action of a powerful ultrasound. A detail of the interface is documented in Figure 14.

EDX analysis was performed to detect the chemical structure and identify the individual phases in the solder joint. The measurement was carried out at five spectrums (Figure 15). The results from this measurement are shown in Table 5.

Spectrums 1 and 2 represent the solder zone formed by the solid solution (Zn) with magnesium.

In the zone of Spectrum 3, the presence of Zn (52 at.%), Si (41 at.%), oxygen (6 at.%), and magnesium (1 at.%) was observed. An interaction between Si and Zn took place. The phase diagram of Si-Zn is of a monotectic type, with a reaction temperature of 420 °C and a monotectic point of 99.95% Zn. Moreover, the interaction of Si and Zn (the Si–Zn system shown in Figure 16) occurred, with an oxide formed on the surface of the ceramic substrate. The results of the measurement prove the interaction between the solder and ceramic material.

In the measurement points of Spectrums 4 and 5, zinc contents of 1 × 22 at.% and 1 × 42 at.% were observed. The silicon content was 21 at.%, while the Mg content varied (11 and 15 at.%), similarly to the oxygen content (46 and 21 at.%). There is thus a complex interaction of all the elements present, in varying proportions. The presence of silicon and magnesium oxides is probable, owing to the higher affinity of these elements to oxygen. The presence of ZnO is less probable, as documented by the analysis of Spectrum 3. If the oxygen were eliminated, it is probable that the structure might be formed by the MgZn_2_ + Mg_2_Si mixture. The mutual interaction of the solder with the ceramics was also proven by the measurement performed on the SiC ceramics/solder boundary and involves the interaction of silicon from the ceramics with magnesium contained in the solder.

The interaction between solder and ceramics is well demonstrated by the planar distribution of the Mg (Figure 17c) and O (Figure 17b) elements on the SiC/Zn3Mg1.5Sr boundary. The magnesium oxides were separated on the joint boundary, which enhanced the bond formation. In spite of the fact that strontium exerts a high affinity to oxygen, no effect of Sr on bond formation was observed.

The line analyses (Figure 18) represent the concentration profile through the SiC/solder boundary for its total length of 9 μm. The interaction of the solder melt with the SiC substrate during soldering occurred on the boundary just partially, in a very narrow layer that was approximately 1 to 2 µm wide. A significant oxygen peak and also a magnesium peak can be observed, which proves their segregation on the ceramics/solder boundary. The interaction of zinc and silicon is also obvious, overlapping for a length of 1.5 µm, which is also proven by the results of the point EDX analysis.

The mechanism of the joint formation can be seen from the results of the point, planar, and line analyses (Figure 19).

The used alloy contains three metals with a high affinity to oxygen. During the soldering process, the magnesium from the solder is distributed to the boundary with the SiC ceramics, where Zn is also present. Due to the effect of soldering in air, the metals oxidize during soldering and are combined with the silicon oxides that occur on the surface of the SiC ceramics. Thus, an interaction based on oxygen takes place while oxygen presents the medium to guarantee the wettability of the SiC ceramics. The resultant mechanical, chemical, and physical properties and the bond strength then attain a standard industrial level.

### 3.6. Analysis of Transition Zone of Cu-SiC/Zn3Mg1.5Sr Joint

The transition zone of the joint (Figure 20) was analyzed. An intense interaction on the Cu-SiC/Zn3Mg1.5Sr joint boundary between the molten zinc solder and the copper matrix of the composite substrate took place at the formation of a wide transition zone formed by the new Cu-Zn phases. The bond formation is thus affected mainly by the Zn from the solder. The thickness of the copper solubility band in the zinc solder is a minimum of 14 µm. The analysis of individual newly-formed phases in the transition zone was performed by EDX. The results of the measurements are given in Table 6.

The locations for the measurements by EDX analysis were selected at five points, namely ranging from Spectrum 1 to Spectrum 5.

At the measurement point of Spectrum 1, a zone of pure copper occurs—a composite matrix with the admixture of Si.

At the measurement point of Spectrum 2, the γCu (Cu_5_Zn_8_) phase was unequivocally defined in agreement with the Cu-Zn binary diagram (the range of occurrence at room temperature varies from 60 to 67 at.% Zn) [23].

Analysis of the zone between the γCu interlayer and solder at the measurement point of Spectrum 3 shows the presence of Zn, Cu, and Mg elements. This involves the two-phase zone of MgCu_2_ + Cu_0.2_Zn_0.8_, and, eventually, MgCu_2_ + ε CuZn_4_ (Figure 21).

At the measurement point of Spectrum 4, the presence of Zn, Cu, and Mg elements with a lower content of copper was also observed. This region, in all probability, concerns the two-phase zone of Mg_2_Zn_11_ + Cu_0.2_Zn_0.8_, and, eventually, Mg_2_Zn_11_ + ε CuZn_4_ (Figure 22).

At the last measurement point of Spectrum 5, an admixture of zinc oxide (ZnO) was identified.

The line analyses in Figure 23 represent the concentration profile through the Cu/solder boundary for a total length of 32 μm. The interaction of molten solder with the Cu substrate occurred and the copper was partly dissolved in the solder due to diffusion at the formation of the new phase—γCu (Cu_5_Zn_8_), which is demarcated by the concentration profile of Cu and Zn shown in Figure 23b, behind which the identified two-phase zone of MgCu_2_ + ε CuZn_4_, 3 to 5 µm wide, follows.

The map of Cu and Zn elements (Figure 24b,c) in the boundary of the Cu-SiC/Zn3Mg1.5Sr joint proves the presence of the γCu (Cu_5_Zn_8_) phase. From the map of the oxygen element (Figure 24f), the presence of ZnO oxides may also be observed on the solder/γCu phase boundary; they were introduced into the solder during ultrasonic soldering in the air. No presence of strontium was noted in the boundary vicinity.

### 3.7. Shear Strength of Soldered Joints

This research study was primarily focused on the soldering of SiC ceramics with the composite substrate of Cu-SiC. Owing to the further feasibility of Zn3Mg1.5Sr solder and its potential application in practice, the shear testing was also extended to other ceramic materials (Al_2_O_3_, AlN, and Si_3_N_4_). The ceramic materials of Al_2_O_3_/Al_2_O_3_, AlN/AlN, and Si_3_N_4_/Si_3_N_4_ were tested and soldered mutually.

A measurement was performed on three specimens of each material. The shear strength results are documented in Figure 25. The highest average shear strength was obtained with the Al_2_O_3_/Al_2_O_3_ joint—up to 106 MPa. In the case of the other joints of the two ceramic materials, namely AlN/AlN and Si_3_N_4_/Si_3_N_4_, similar shear strength values of 99.5 and 93 MPa were observed.

In the case of the SiC/Cu-SiC joint, a slightly lower shear strength of 62 MPa was achieved, but this still meets the condition for demanding industrial applications. The limit value of shear strength for the applications of power electronic modules is 40 MPa. This value was greatly exceeded by the application of Zn3Mg1.5Sr solder, showing, therefore, that it is an excellent candidate for such applications.

The mentioned results show that the strength values of joints of similar ceramic materials are comparable at the value of around 100 MPa. However, in the case of combined joints, their shear strength drops by 40%, owing to residual stresses caused by the different thermal expansivity of SiC ceramics and Cu-SiC composite.

For a more specific identification of the mechanism of bond creation, the fractured surfaces of the joints were analyzed. Figure 26a,b shows the documented fractured surfaces from the boundary of the SiC/Zn3Mg1.5Sr/Cu-SiC joint. It is obvious that the fractured surface from the side of the SiC ceramics stayed almost completely covered with solder. In addition, SiC grains may be observed locally on the fractured surface. A ductile fracture has occurred in the solder. Analysis of the planar distribution of Mg, Si, Cu, Zn, and Sr elements on the fractured surface was also performed and is documented in Figure 27b–f. The planar distribution of magnesium, shown in Figure 27b, suggests that magnesium has segregated in the solder/ceramics boundary and locally surrounds the Si grains (Figure 27c), which represent the SiC ceramics. This fact proves that Mg contributes significantly to bond formation. Strontium (Figure 27f) is uniformly distributed on the fractured surface; however, nothing suggests that it would contribute to bond formation.

For that reason, an XRD analysis of the fractured surface from the SiC/Zn3Mg1.5Sr joint boundary was performed (Figure 28) and it proved the facts found. The interaction of Mg from the solder with the surface of the ceramic SiC material at the formation of the new Mg_2_Si phase was proven unequivocally. Similarly, the interaction of Zn with the surface of the Cu-SiC substrate at the formation of Cu_5_Zn_8_ and CuZn_4_ phases was proven. Moreover, the existence of Zn, Mg, and Sr was proven. The MgZn_2_, Mg_2_Zn_11_, and SrZn_13_ phases were identified.

## 4. Conclusions

The aim of the research was to characterize the soldering alloy type Zn-Mg-Sr and to study whether the proposed composition of the soldering alloy was suitable for soldering SiC ceramics and composite material of Cu-SiC with the application of ultrasonic soldering. The following results were achieved:For the determination of the solder melting point, TG/DTA analysis was applied. The record may show a significant peak, which corresponds to a eutectic reaction. The Zn– Mg system is thus of a eutectic type, with a reaction temperature of 364 °C. The effect of strontium on phase transformations was minimal, owing to its lower content.The microstructure of the soldering alloy type Zn3Mg1.5Sr is formed of a very fine eutectic matrix, where the strontium phases (light-grey constituents) and magnesium phases (grey constituents) are segregated. The XRD analysis of the solder revealed the solid solution of zinc (Zn), intermetallic phases of magnesium—MgZn_2_, Mg_2_Zn_11_, and the presence of an intermetallic phase of strontium—SrZn_13_.The solder type Zn3Mg1.5Sr has an average tensile strength of 98.6 MPa. The tensile strength is partially increased by solder alloying with magnesium and strontium.The SiC/solder bond is formed due to the distribution of magnesium from the solder to the boundary with SiC ceramics at the formation of the Mg_2_Si phase. Due to the effect of soldering in air, the metals oxidize during soldering and the oxides formed are combined with the silicon oxides that occur on the surface of SiC ceramics. An interaction based on oxygen as a medium takes place, thus ensuring the wettability of SiC ceramics.A massive interaction on the boundary of the Cu-SiC/solder joint took place between the molten zinc solder and copper matrix of the composite substrate at the formation of a wide transition zone, formed mainly of the new phase of γCu (Cu_5_Zn_8_). The bond formation is thus affected mainly by the Zn from the solder.The shear strength was measured using several ceramic materials. The average shear strength of the combined joint of SiC/Cu-SiC fabricated by using Zn3Mg1.5Sr solder was 62 MPa. When soldering similar ceramic materials mutually, a shear strength of as much as around 100 MPa was observed. The limit value of shear strength for the application of power electronic modules is 40 MPa. The application of Zn3Mg1.5Sr solder greatly exceeded this value and it is therefore an excellent candidate for such applications.

## Figures and Tables

**Figure 1 materials-16-03795-f001:**
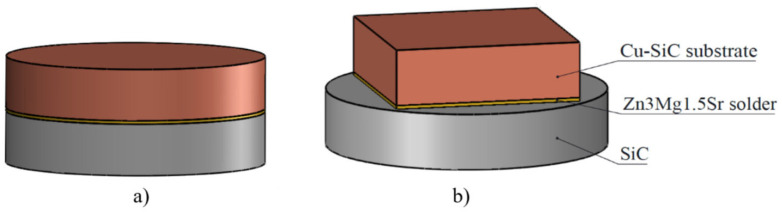
Formation of soldered joint: (**a**) joint used for solder/substrate interface analysis, (**b**) joint used for shear strength test.

**Figure 2 materials-16-03795-f002:**
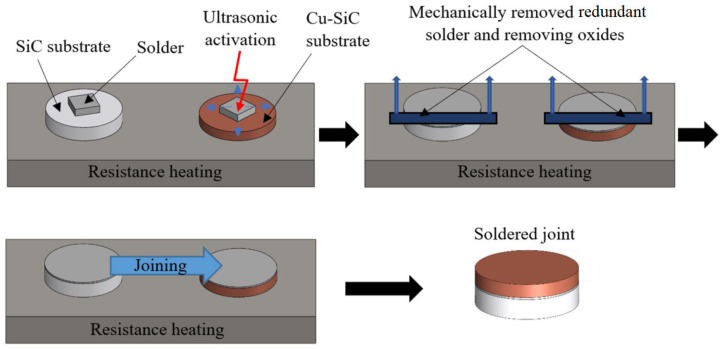
Schematic representation of the soldering process in the presence of ultrasonic energy.

**Figure 3 materials-16-03795-f003:**
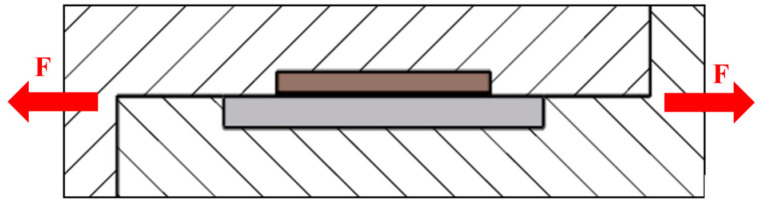
Shear strength measurement scheme.

**Figure 4 materials-16-03795-f004:**
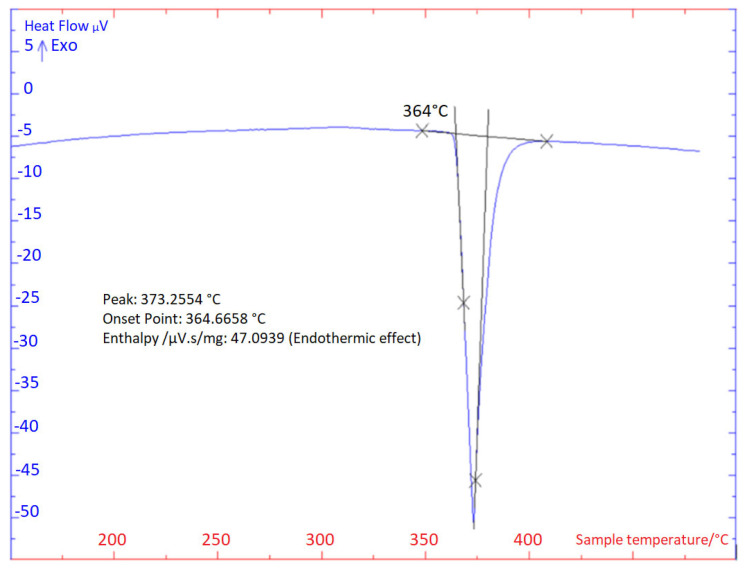
TG/DTA solder analysis of Zn3Mg1.5Sr heating at 5 °C/min, second heating.

**Figure 5 materials-16-03795-f005:**
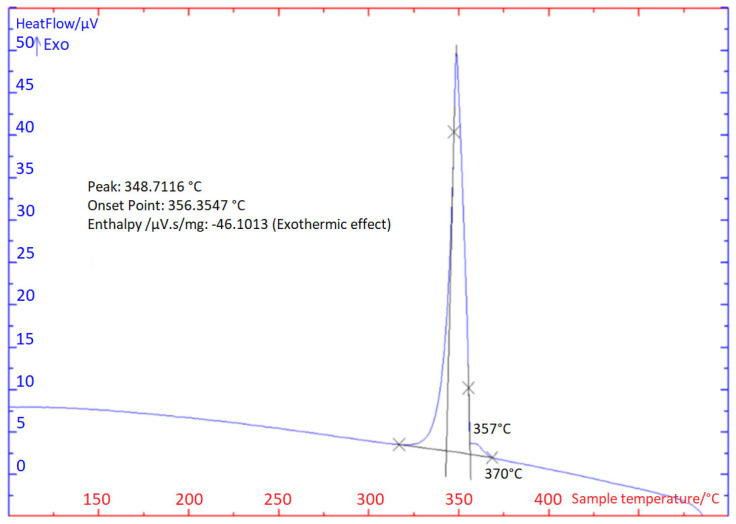
TG/DTA solder analysis of Zn3Mg1.5Sr cooling at 5 °C/min, second cooling.

**Figure 6 materials-16-03795-f006:**
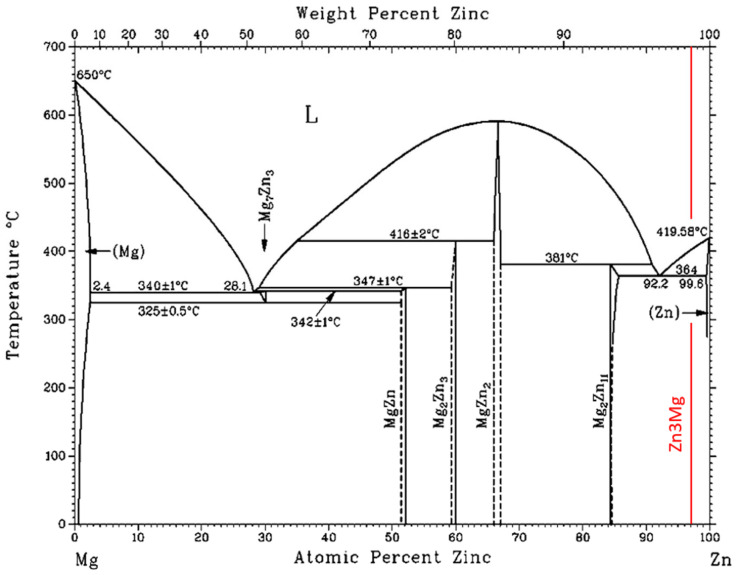
Equilibrium binary diagram of magnesium-zinc system (Re-printed with permission from ASM International) [26].

**Figure 7 materials-16-03795-f007:**
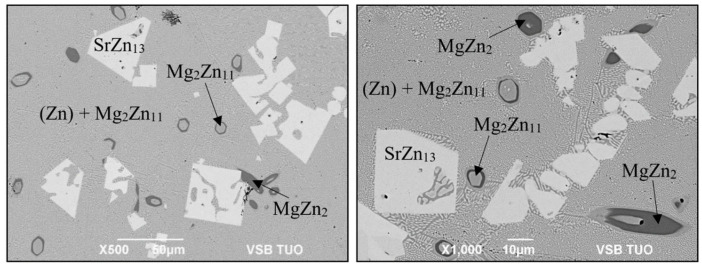
Microstructure of Zn3Mg1.5Sr solder from SEM (BEI) analysis in as-etched condition.

**Figure 8 materials-16-03795-f008:**
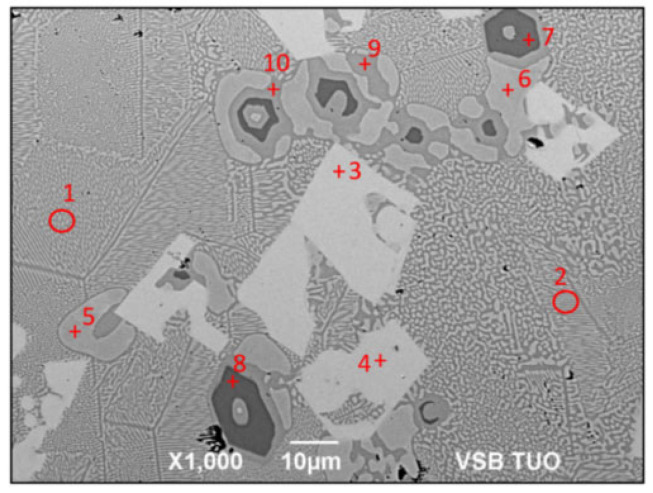
Point energy dispersive X-ray analysis of Zn3Mg1.5Sr solder.

**Figure 9 materials-16-03795-f009:**
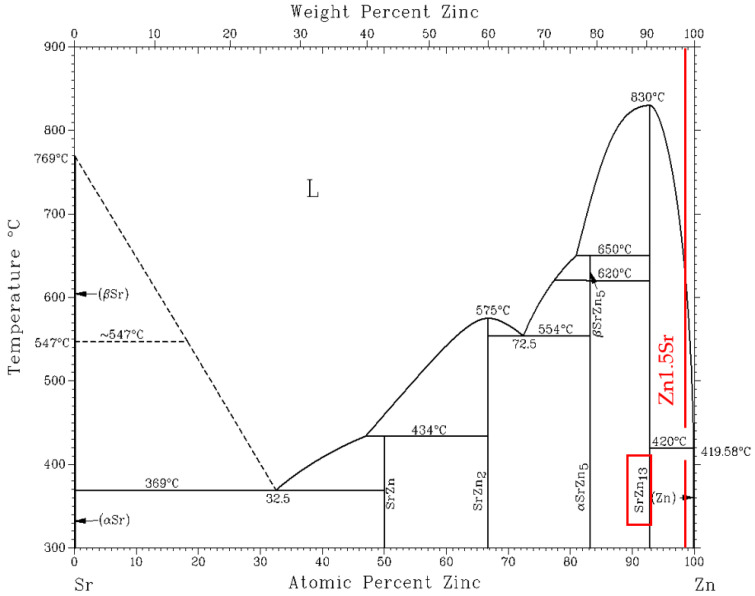
Equilibrium binary diagram of strontium–zinc system (Re-printed with permission from ASM International) [26].

**Figure 10 materials-16-03795-f010:**
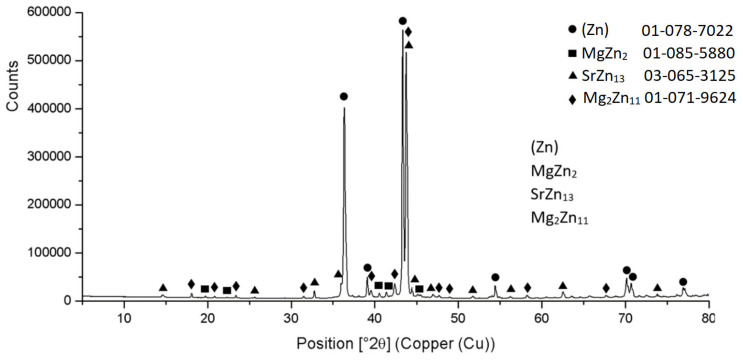
XRD analysis of solder Zn3Mg1.5Sr.

**Figure 11 materials-16-03795-f011:**
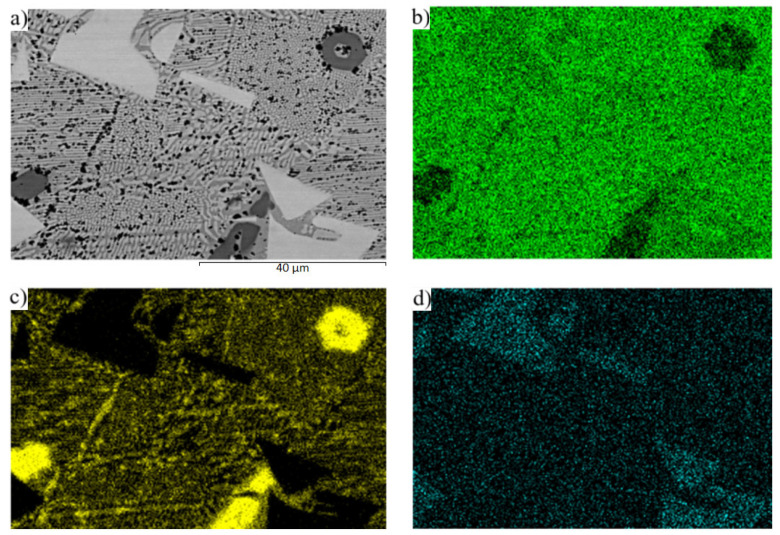
Map of (**a**) microstructure solder, (**b**) Zn, (**c**) Mg, and (**d**) Sr elements in the microstructure of Zn3Mg1.5Sr solder.

**Figure 12 materials-16-03795-f012:**
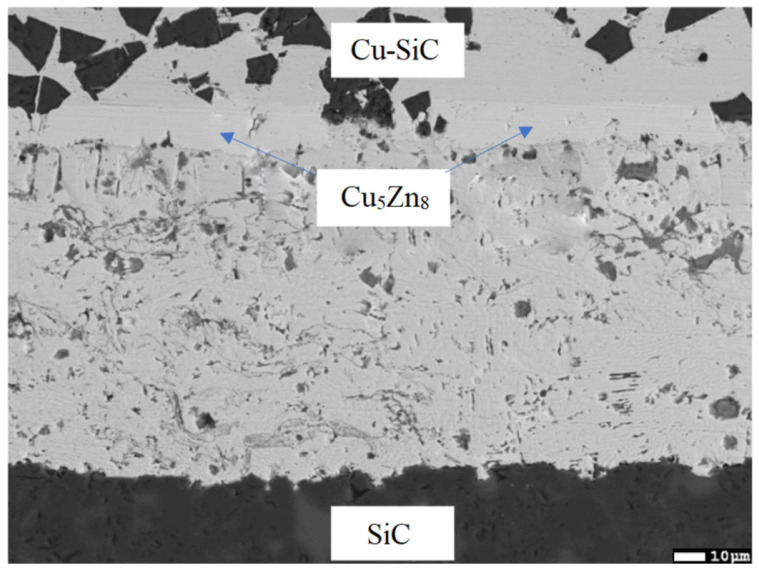
Microstructure of boundary in SiC/Zn3Mg1.5Sr/Cu-SiC joint from the SEM analysis performed in BSE regime.

**Figure 13 materials-16-03795-f013:**
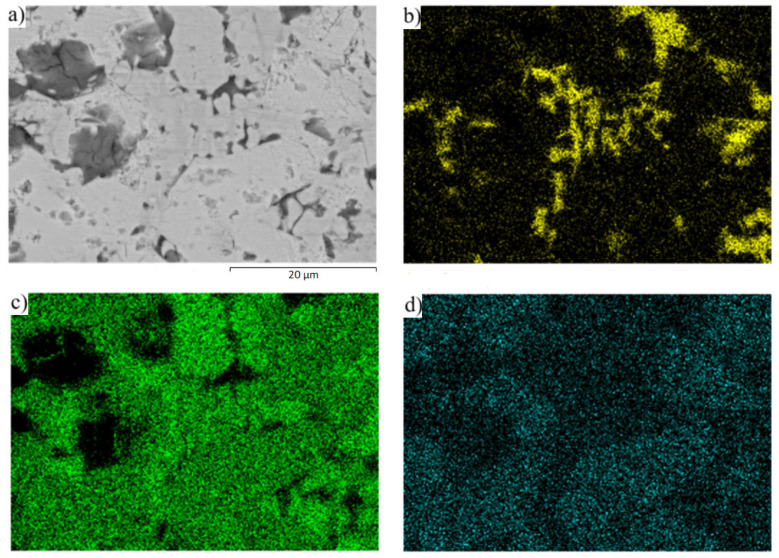
Map of (**a**) microstructure solder, (**b**) Mg, (**c**) Zn, and (**d**) Sr elements in the microstructure of solder after soldering of SiC/Zn3Mg1.5Sr/Cu-SiC joint.

**Figure 14 materials-16-03795-f014:**
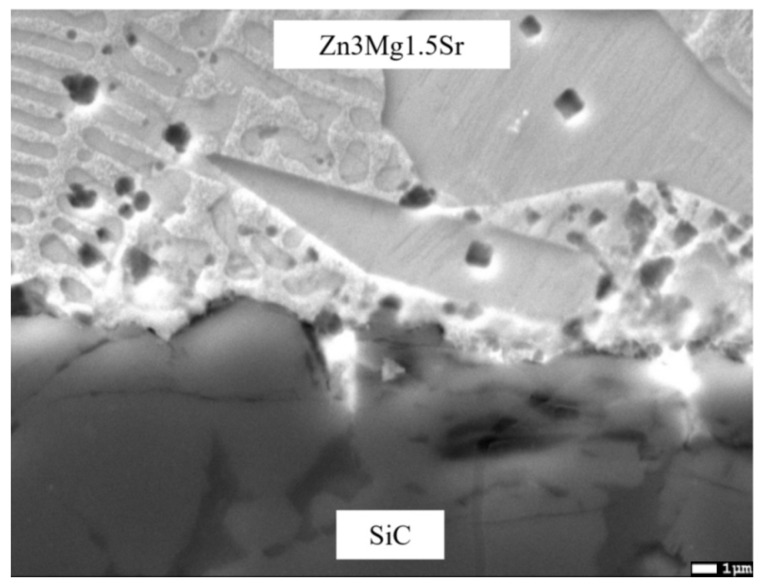
Microstructure of transition zone in SiC/Zn3Mg1.5Sr joint.

**Figure 15 materials-16-03795-f015:**
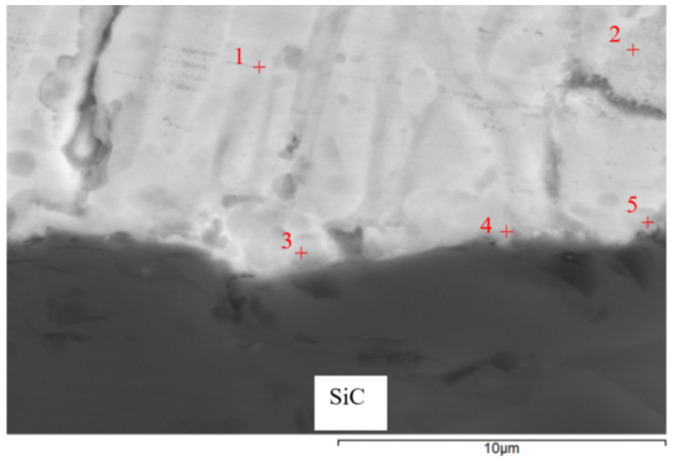
Point energy dispersive X-ray analysis of SiC/Zn3Mg1.5Sr joint.

**Figure 16 materials-16-03795-f016:**
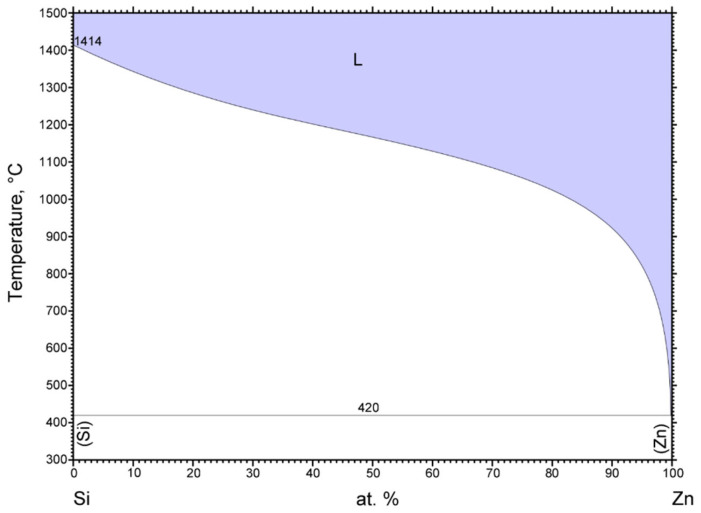
Equilibrium binary diagram of silicon–zinc system (Re-printed with permission from ASM International) [26].

**Figure 17 materials-16-03795-f017:**
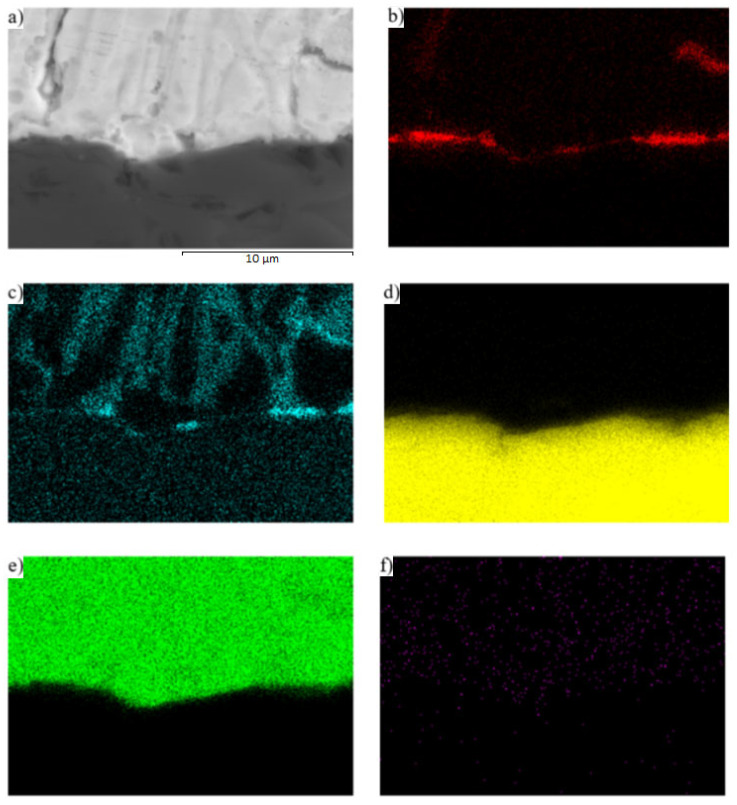
Planar distribution of (**a**) joint microstructure, (**b**) O, (**c**) Mg, (**d**) Si, (**e**) Zn, and (**f**) Sr elements on the boundary of SiC/Zn3Mg1.5Sr joint.

**Figure 18 materials-16-03795-f018:**
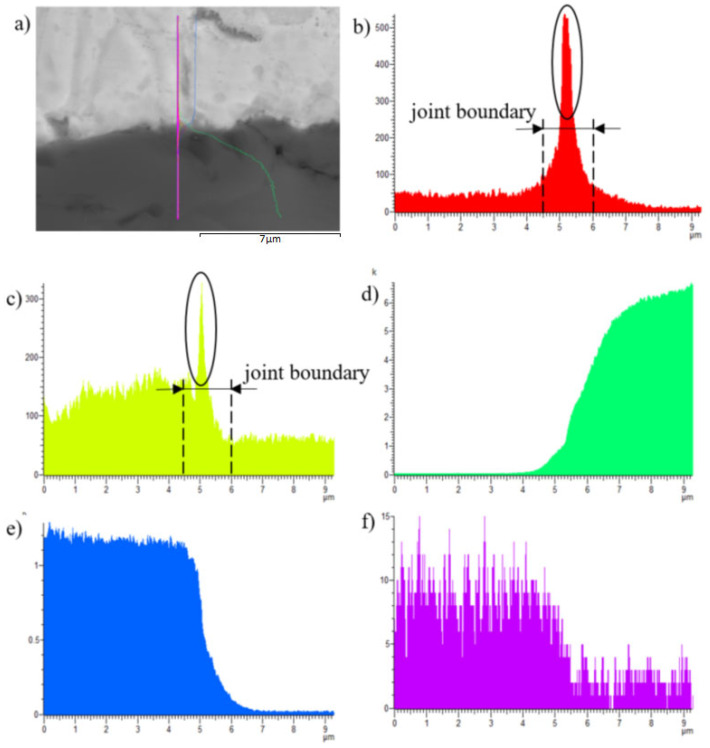
Line EDX analysis of Si/Zn3Mg1.5Sr joint. Transition zone with a marked line (**a**); (**b**) oxygen; (**c**) magnesium; (**d**) silicon; (**e**) zinc; (**f**) strontium; concentration profiles of elements.

**Figure 19 materials-16-03795-f019:**
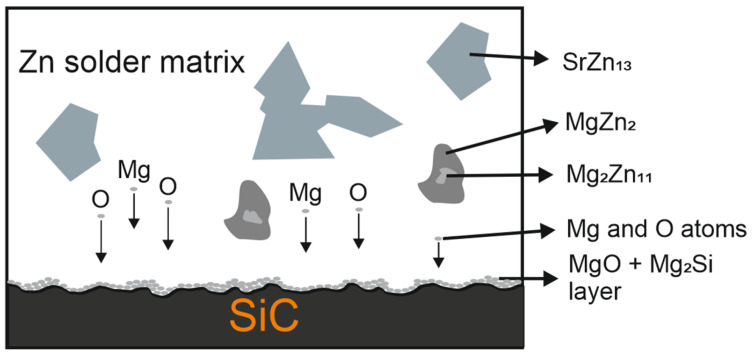
Mechanism of the joint formation.

**Figure 20 materials-16-03795-f020:**
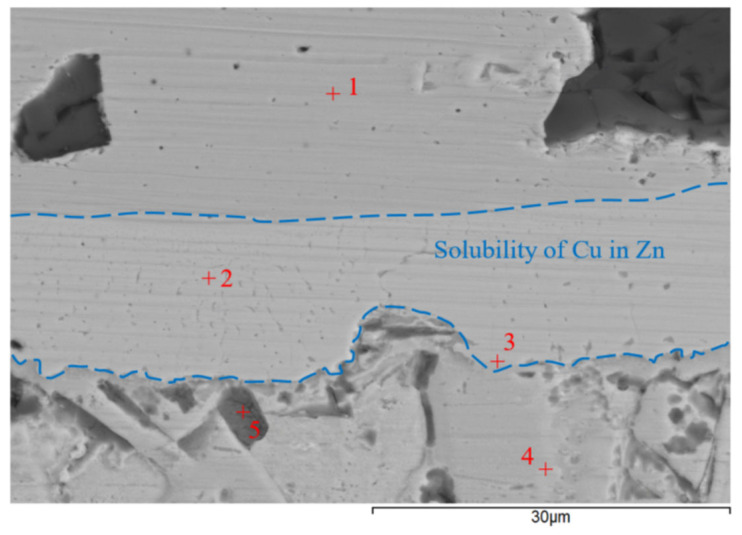
Point energy dispersive X-ray analysis of Cu-SiC/Zn3Mg1.5Sr joint.

**Figure 21 materials-16-03795-f021:**
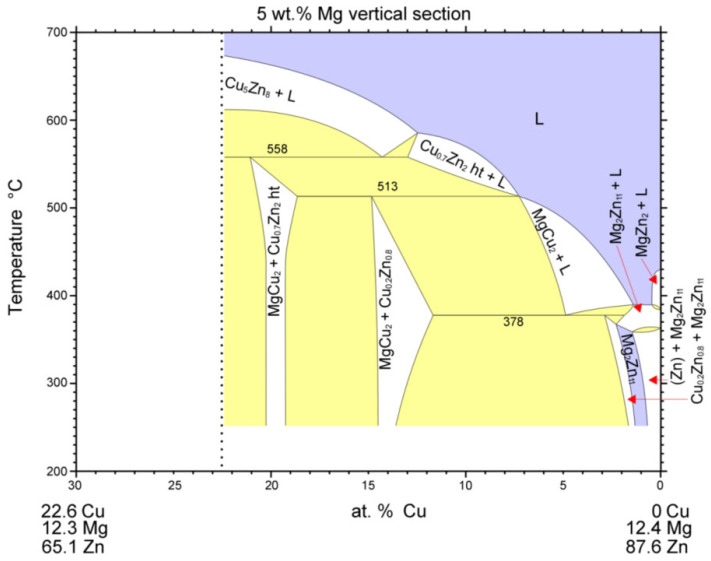
Isopletic cross-section through the Cu-Mg-Zn ternary system for 5 wt. % Mg (Re-printed with permission from Elsevier) [27].

**Figure 22 materials-16-03795-f022:**
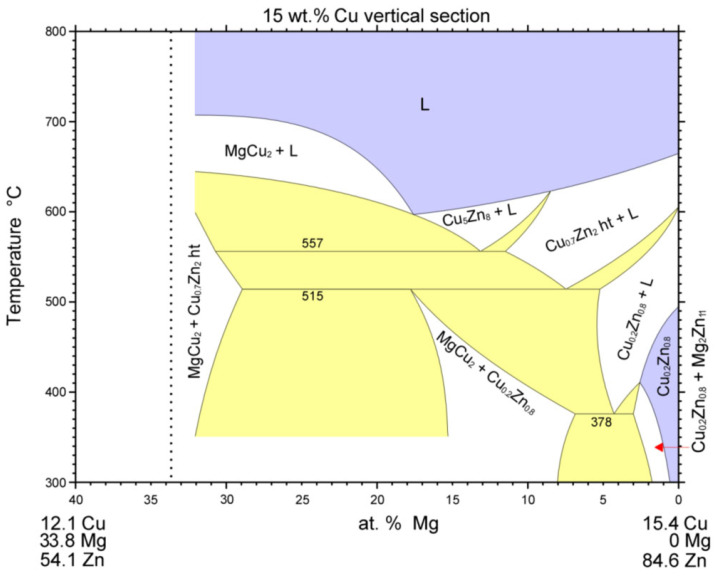
Isopletic cross-section through the Cu-Mg-Zn ternary system for 15 wt. % Mg (Re-printed with permission from Elsevier) [27].

**Figure 23 materials-16-03795-f023:**
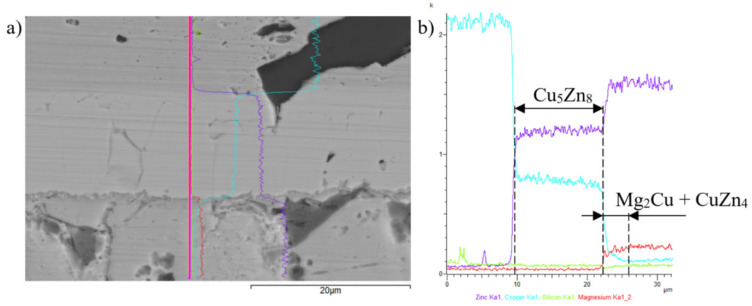
Line EDX analysis of Cu-SiC/Zn3Mg1.5Sr joint: (**a**) the transition zone with a marked line, (**b**) concentration profiles of Zn, Cu, Si, and Mg elements.

**Figure 24 materials-16-03795-f024:**
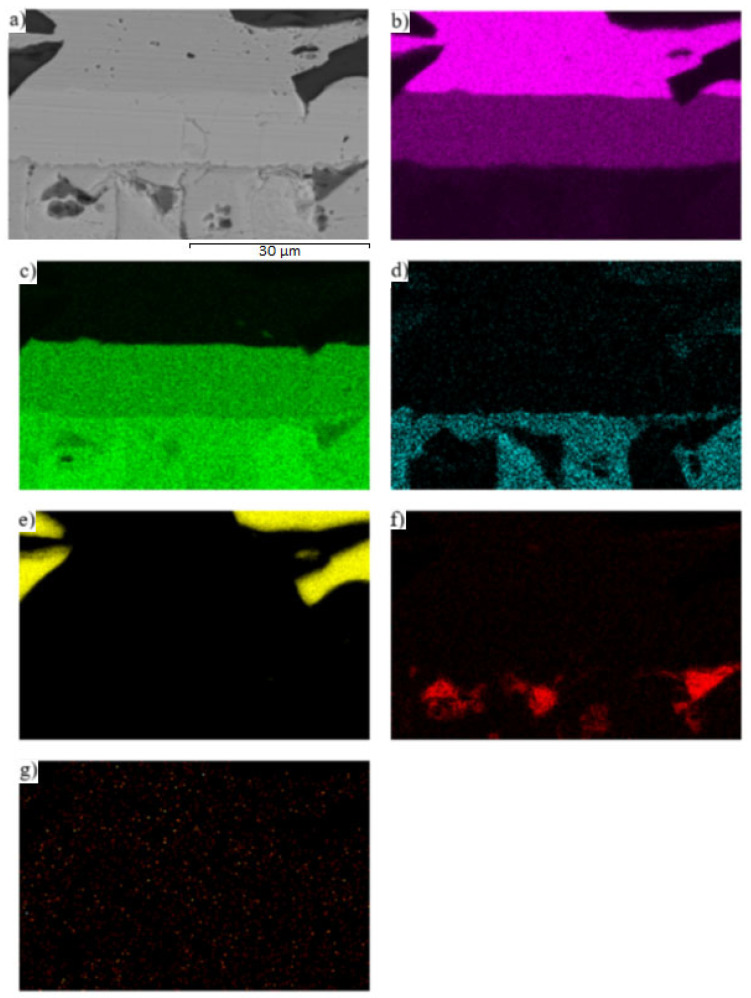
The map of (**a**) joint microstructure, (**b**) Cu, (**c**) Zn, (**d**) Mg, (**e**) Si, (**f**) O, and (**g**) Sr elements on the boundary of Cu-SiC/Zn3Mg1.5Sr joint.

**Figure 25 materials-16-03795-f025:**
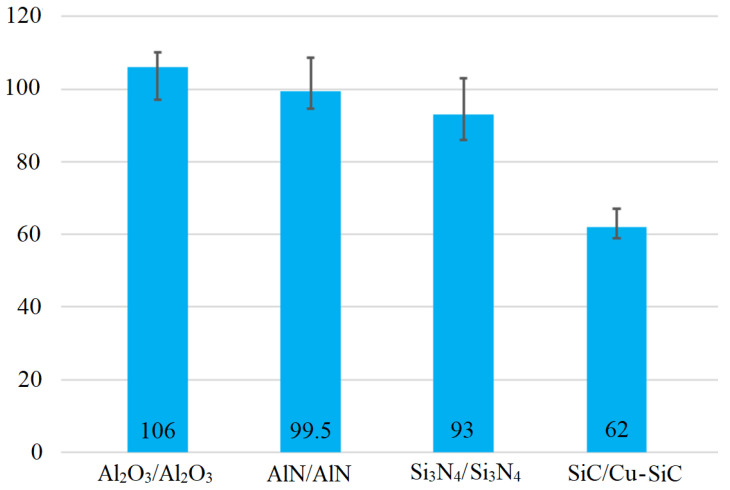
Shear strength of soldered joints fabricated by use of Zn3Mg1.5Sr solder.

**Figure 26 materials-16-03795-f026:**
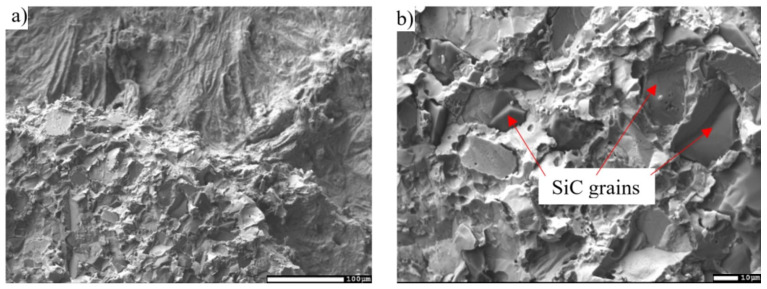
(**a**) Fractured surface of SiC/Zn3Mg1.5Sr/Cu-SiC joint; (**b**) Fractured surface of SiC/Zn3Mg1.5Sr/Cu-SiC joint with higher magnification.

**Figure 27 materials-16-03795-f027:**
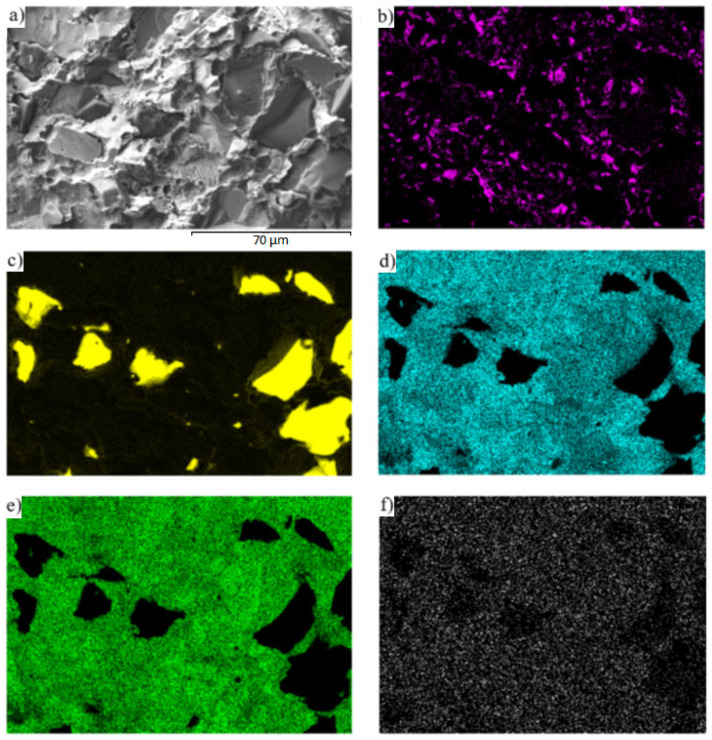
Fractured surface of SiC/Zn3Mg1.5Sr/Cu-SiC joint and the planar distribution of individual elements: (**a**) fracture structure, (**b**) Mg, (**c**) Si, (**d**) Cu, (**e**) Zn, (**f**) Sr.

**Figure 28 materials-16-03795-f028:**
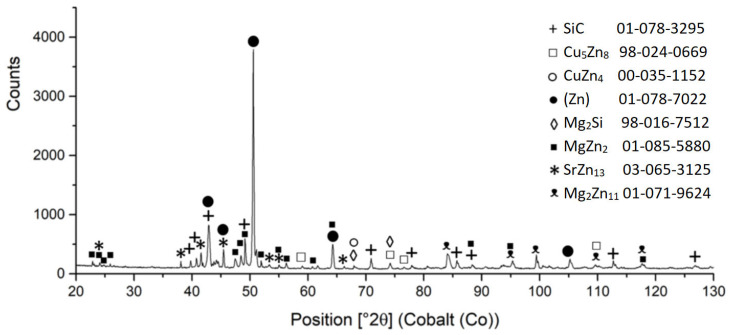
XRD analysis of fractured surface from SiC/Zn3Mg1.5Sr joint boundary.

**Table 1 materials-16-03795-t001:** Composition of Zn-Mg-Sr solder and ICP-AES chemical analysis results [wt. %].

Specimen	Charge [wt. %]			ICP-AES [wt. %]		
	Zn	Mg	Sr	Zn	Mg	Sr
**Zn3Mg1.5Sr**	95.5	3	1.5	96.04	2.84	1.12

**Table 2 materials-16-03795-t002:** Soldering parameters.

**Ultrasound power**	400	[W]
**Working frequency**	40	[kHz]
**Amplitude**	2	[μm]
**Soldering temperature**	150	[°C]
**Time of ultrasound activation**	5	[s]

**Table 3 materials-16-03795-t003:** Important temperatures of phase transformations achieved by TG/DTA analysis.

DTA Analysis		*T_L_* (°C)	*T_E_* (°C)
**Heating**	1st	368	365
	2nd	368	364
**Cooling**	1st	370	356
	2nd	370	357

T_L_—Temperature of liquidus, T_E_—Temperature of eutectic transformation.

**Table 4 materials-16-03795-t004:** Results from energy dispersive X-ray analysis on Zn3Mg1.5Sr solder.

Spectrum	Mg [at.%]	Zn [at.%]	Sr [at.%]	Solder Component
**Spectrum 1**	8.41	91.59	-	Eutectic (Zn) + Mg_2_Zn_11_
**Spectrum 2**	9.13	90.87	-	Eutectic (Zn) + Mg_2_Zn_11_
**Spectrum 3**	-	92.81	7.19	SrZn_13_ phase
**Spectrum 4**	-	92.81	7.19	SrZn_13_ phase
**Spectrum 5**	-	100.00	-	Zn matrix
**Spectrum 6**	-	100.00	-	Zn matrix
**Spectrum 7**	34.05	65.95	-	MgZn_2_ phase
**Spectrum 8**	33.71	66.29	-	MgZn_2_ phase
**Spectrum 9**	16.61	83.39	-	Mg_2_Zn_11_ phase
**Spectrum 10**	16.19	83.81	-	Mg_2_Zn_11_ phase

**Table 5 materials-16-03795-t005:** The results from point energy dispersive X-ray analysis of SiC/Zn3Mg1.5Sr joint.

Spectrum	Zn [at.%]	Mg [at.%]	Si [at.%]	O [at.%]
**Spectrum 1**	97.62	2.38	0	0
**Spectrum 2**	91.78	8.22	0	0
**Spectrum 3**	51.75	1.00	41.28	5.97
**Spectrum 4**	21.83	11.16	21.36	45.65
**Spectrum 5**	42.42	15.33	21.13	21.12

**Table 6 materials-16-03795-t006:** The results from point energy dispersive X-ray analysis of Cu-SiC/Zn3Mg1.5Sr joint.

Spectrum	Zn [at.%]	Mg [at.%]	Si [at.%]	O [at.%]	Cu [at.%]
**Spectrum 1**	0	0	1.68	0	98.32
**Spectrum 2**	65.61	0	1.06	0	33.33
**Spectrum 3**	73.45	14.98	0	0	11.57
**Spectrum 4**	78.35	18.73	0	0	2.92
**Spectrum 5**	45.60	0.95	0	52.09	1.36

## Data Availability

The data given in this study are available on request from the appropriate author.

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
