# Peer review of "Characterization of Zn-Mg-Sr Type Soldering Alloy and Study of Ultrasonic Soldering of SiC Ceramics and Cu-SiC Composite"

_materials, 2023, doi:10.3390/ma16103795_

Round 1

Reviewer 1 Report

In my opinion, the current status of this manuscript cannot be accepted, and the main issues are as follows:

1. the English  grammar should be carefully revised.

2. The images are blurry and should be re-uploaded.

3. In Figure 13, the interfacial reaction layer shows a distinct single-phase structure, why are there two phases marked?

4. It is necessary to conduct transmission electron microscopy analysis on the interfacial reaction phase or brazing seam layer to clarify the phase composition.

Author Response

  1. The English language has been proofread by Proof-Reading-Service.com, I enclose a proofreading certificate.
  2. The quality of images has been adjusted.
  3. The incorrect phases were shown in the figure, there is Cu5Zn8 phase in the reaction layer.
  4. Performing analyses by transmission electron microscopy is currently not possible due to time constraints, as the editor has only given a 10-day deadline to revise the article after receipt of the reviews.

Reviewer 2 Report

The article entitled "Characterization on Zn-Mg-Sr type soldering alloy and study of ultrasonic soldering of SiC ceramics and Cu-SiC composite" is focused on the characterization of the soldering alloy type Zn-Mg-Sr and direct soldering of SiC ceramics with Cu-SiC based composite.

The presented research is interesting and sufficiently novel. The scope of the manuscript is original, but I have some remarks.

·     The authors in their previous papers, for example,

- R. Koleňák, I. Kostolný, J. Drápala, M. Sahul, J. Urminský, Characterizing the Soldering Alloy Type In–Ag–Ti and the Study of Direct Soldering of SiC Ceramics and Copper, Metals 8(4), 2018, 274‒274 and

- R. Koleňák, I. Kostolný, J. Drápala, P. Babincová, [P. Zacková], P. Gogola, Characterization of Soldering Alloy Type Bi-Ag-Ti and the Study of Ultrasonic Soldering of Silicon and Copper, Metals 11(4), 2021, 1‒21.

use the same figures for Test piece of solder for the static tensile test, Schematic representation of the soldering process, and Shear strength measurement scheme. It should be avoided and refer to the already published papers.

·         the quality of some diagrams is poor (Figures 5, 6)

·         In figure 9 semi-quantities analysis was given in wt.%, and in figure 16, and 21 in at.%. Explain the reason.

Author Response

  1. Thank you for your comment/advice, it will be incorporated in future publications.
  2. The quality of images has been adjusted.
  3. Wt.% have been converted to at.% and the table has been adjusted.

Reviewer 3 Report

The article "Characterization on Zn-Mg-Sr type soldering alloy and study of ultrasonic soldering of SiC ceramics and Cu-SiC composite" is suitable for publication in Materials Journal after some major corrections. The paper is genenrally satisfacatory written but there need some improvements:

1. The abstract should contain in the first part some introductive aspects of the topic.

2. The introduction should state some aspects regarding the influence of the chemical elements in terms of corrosion and mechanical properties, especially Zn content. Suggested references:10.3390/cryst12101468, 10.3390/app12052727.

3. Please explain more about the parameters for SEM and XRD analysis.

4. Figures 5 and 6 are not very clear. "The TG/DTA analysis was carried out two times in each case at the heating and cooling rate of 5 °C/min (Figs. 5 and 6)." More explanations need to be added.

5. figure 9 should be split in two: table and figure, also not very clear the points for eds analysis.

6. For Figure 11-XRD analysis, please add the ICDD files for the compound and corelate with the SEM images.

7. figure 16&21 . split in two

8. In my opinion, all the binary diagrams should be in a single figure at the beginning of the results.

9. Improve the quality of the figure 26.

10. There are too many figures; please merge some of them.

The rest is fine.

Author Response

  1. The abstract has been supplemented.
  2. The introduction has been supplemented with recommended studies.
  3. Parameters for SEM/EDX and XRD analyses have been added.
  4. The quality of images has been adjusted.
  5. Figure 9 was divided into two parts.
  6. Figures 11 and 29 have been supplemented with numbered phase labels from the ICDD files.

SiC         01-078-3295

Cu5Zn8    98-024-0669

CuZn4    00-035-1152

(Zn)        01-078-7022

Mg2Si     98-016-7512

MgZn2    01-085-5880

SrZn13     03-065-3125

Mg2Zn11 01-071-9624

  1. Figures 16 and 21 were divided into two parts.
  2. Thank you for your comment, in future publications we will try to group all binary diagrams on top of the results.
  3. The quality of image 26 was adjusted.
  4. Thank you for your comment, but the pictures in the article have their own merits and therefore I think it is both inappropriate and not possible to merge them in some way.

Round 2

Reviewer 1 Report

The authors has revised the manuscript accordingly.

Author Response

Thank you for your substantive comments on our article.

Reviewer 2 Report

Dear colleagues,

My final desition is that this manuscript should be accepted in this form.

Best regards

Author Response

(The authors gave the same response as above.)
